# Mature MUC5AC Expression in Resected Pancreatic Ductal Adenocarcinoma Predicts Treatment Response and Outcomes

**DOI:** 10.3390/ijms25169041

**Published:** 2024-08-20

**Authors:** Ashish Manne, Ashwini Esnakula, Ankur Sheel, Amir Sara, Upender Manne, Ravi Kumar Paluri, Kai He, Wancai Yang, Davendra Sohal, Anup Kasi, Anne M. Noonan, Arjun Mittra, John Hays, Sameek Roychowdhury, Pannaga Malalur, Shafia Rahman, Ning Jin, Jordan M. Cloyd, Susan Tsai, Aslam Ejaz, Kenneth Pitter, Eric Miller, Kannan Thanikachalam, Mary Dillhoff, Lianbo Yu

**Affiliations:** 1Department of Internal Medicine, Division of Medical Oncology, The Ohio State University Comprehensive Cancer Center (OSU-CCC), Columbus, OH 43210, USA; 2Department of Pathology, The Ohio State University Comprehensive Cancer Center (OSU-CCC), Columbus, OH 43210, USA; ashwini.esnakula@osumc.edu; 3Department of Pathology, University of Alabama at Birmingham, Birmingham, AL 35233, USA; 4Division of Hematology-Oncology, Department of Internal Medicine, Atrium Health Wake Forest Baptist Comprehensive Cancer Center, Winston-Salem, NC 27103, USA; 5Comprehensive Cancer Center, The Ohio State University, Columbus, OH 43210, USA; 6Department of Internal Medicine, Division of Hematology/Oncology, College of Medicine, University of Cincinnati, Cincinnati, OH 45267, USA; 7Division of Medical Oncology, University of Kansas Cancer Center, Westwood, KS 66205, USA; 8Department of Surgery, Division of Surgical Oncology, The Ohio State University Comprehensive Cancer Center (OSU-CCC), Columbus, OH 43221, USA; 9Department of Surgical Oncology, University of Illinois College of Medicine, Chicago, IL 60612, USA; 10Department of Radiation Oncology, The Ohio State University Comprehensive Cancer Center (OSU-CCC), Columbus, OH 43210, USAeric.miller@osumc.edu (E.M.); 11Center of Biostatistics and Bioinformatics, Roswell Park Comprehensive Cancer Center, 665 Elm St, Buffalo, NY 14203, USA; 12Center of Biostatistics and Bioinformatics, The Ohio State University, Columbus, OH 43210, USA

**Keywords:** pancreatic adenocarcinoma, MUC5AC, neoadjuvant therapy, prognosis, recurrence, resectable pancreatic cancer, predictive biomarker

## Abstract

Neoadjuvant therapy (NAT) for early-stage pancreatic ductal adenocarcinoma (PDA) has recently gained prominence. We investigated the clinical significance of mucin 5 AC (MUC5AC), which exists in two major glycoforms, a less-glycosylated immature isoform (IM) and a heavily glycosylated mature isoform (MM), as a biomarker in resected PDA. Immunohistochemistry was performed on 100 resected PDAs to evaluate the expression of the IM and MM of MUC5AC using their respective monoclonal antibodies, CLH2 (NBP2-44455) and 45M1 (ab3649). MUC5AC localization (cytoplasmic, apical, and extra-cellular (EC)) was determined, and the H-scores were calculated. Univariate and multivariate (MVA) Cox regression models were used to estimate progression-free survival (PFS) and overall survival (OS). Of 100 resected PDA patients, 43 received NAT, and 57 were treatment-naïve with upfront surgery (UpS). In the study population (*n* = 100), IM expression (H-scores for objective response vs. no response vs. UpS = 104 vs. 152 vs. 163, *p* = 0.01) and MM-MUC5AC detection rates (56% vs. 63% vs. 82%, *p* = 0.02) were significantly different. In the NAT group, MM-MUC5AC-negative patients had significantly better PFS according to the MVA (Hazard Ratio: 0.2, 95% CI: 0.059–0.766, *p* = 0.01). Similar results were noted in a FOLFIRINOX sub-group (*n* = 36). We established an association of MUC5AC expression with treatment response and outcomes.

## 1. Introduction

Pancreatic ductal adenocarcinoma (PDA) is one of the deadliest cancers, with more than 66,000 cancers diagnosed every year and the lowest 5-year survival rate (13%) among solid tumors [1]. Limited treatment options and unreliable biomarkers to monitor response are the main reasons for such dismal outcomes, which we discussed in our previous publication [2]. Until a decade ago, the management of resected PDA (R-PDA) was straightforward, with upfront surgery (UpS) followed by adjuvant chemotherapy (AT) with FOLFIRINOX or the gemcitabine/capecitabine (Gem/Cap) combination [3,4,5]. Retrospective analyses and data from a randomized phase II/III trial, Prep-02/JSAP-05, support the use of neoadjuvant systemic therapy for resectable pancreatic cancers. However, long-term data and more randomized clinical trials are warranted [6].

Neoadjuvant therapy (NAT) for PDA has gained traction in the last 5–6 years and is now preferred in most institutions [7,8,9,10,11,12]. Patient selection beyond the stage of diagnosis (resectable (Rs-PDA) vs. borderline resectable (BRs-PDA) vs. locally advanced (LA-PDA)), the modality used (chemoradiation (CRT) vs. chemotherapy vs. both), the chemotherapy agents (Gem vs. FOLFIRINOX vs. Gem/nab-paclitaxel (Gem-NP)), and the duration of perioperative therapy that would make a meaningful difference in the outcomes are still unclear [13,14,15].

The biomarkers applied in current clinical practice, including imaging, carbohydrate antigen 19-9 (CA 19-9), and cell-free DNA (cfDNA) testing, are not useful in making appropriate treatment decisions [2,16,17]. Our group previously focused on tissue mucin 5 AC (MUC5AC) as a biomarker for managing PDA and biliary tract cancer [18,19,20,21,22]. In prior publications, we assessed its diagnostic value and provided preclinical evidence suggesting its influence on the response to systemic therapy [19,21]. The present effort examined the role of MUC5AC in the outcomes of patients undergoing curative resection (R-PDA). 

MUC5AC is a heavy glycoprotein typically produced in normal lungs and gastrointestinal tracts along with other mucins to protect them from infection, inflammation, and other physiological insults [23,24]. Its presence in pancreatic tissues is abnormal and is often associated with malignancy [21,25,26,27,28,29,30]. In PDA cells, MUC5AC exists in two glycoforms: the less-glycosylated immature (IM) detected in the perinuclear area, and the heavily glycosylated mature form (MM) detected in the apical and extracellular (EC) regions [28]. Commercially available monoclonal antibodies (mAb) allow us to distinguish them in tissues via immunohistochemistry (IHC). The CLH2 clone detects IM, and MM is identified by 45M1, 1-13M1, 9-13, and 2-11M1 mAbs [31,32,33]. Prior studies that examined the influence of IM on PDAs were for UpS patients or a mixed population (early- and advanced-stage PDA), and the results were inconclusive [34,35,36,37,38]. The clinical significance of MM has not been investigated well despite preclinical evidence suggesting that it has prognostic and predictive value [39]. 

We believed we had been considering the wrong MUC5AC glycoform (IM) to investigate PDA outcomes or treatment resistance. We hypothesized that the MM glycoform, which is detected in apical and EC sites in PDA, affects the overall outcome and treatment resistance (prognostic and predictive value) based on the preclinical evidence. Our present study explored the clinical significance of IM and MM in resected PDA specimens (R-PDA). 

## 2. Results 

### 2.1. Baseline Characteristics

We received 112 tumor samples (see Appendix A). However, samples for 12 patients had to be excluded (1 had microscopic liver metastasis; 9 were samples from metastatic sites such as the lung, liver, omentum, ovary, and bladder; 1 had a neuroendocrine pancreatic tumor; and the other patient’s slides did not have tumor tissue even though there was a residual tumor in the resected sample (according to the pathology report)). We ended up with 100 patients in the study who had curative resection for PDA. 

The median age of diagnosis was 65 years, with an equal number of male and female patients. A documented history of pancreatic cysts or IPMN (on imaging) before surgery was noted for 20%, and all had diabetes mellitus (DM). A total of 43 patients had NAT (36 FOLFIRINOX, 5 Gem-NP, and 2 FOLFOX); the other 57 had UpS. Sixteen (16) patients had neoadjuvant chemoradiation (NAT-CRT) post-NAT. Among the UpS patients, 55/57 received systemic adjuvant therapy (3 FOLFIRINOX, 7 Gem-NP, 33 Gem-only, 11 Gem/Cap, 1 Cap-only). Two patients had adjuvant CRT only (one with Gem and the other with 5FU). The median number of NAT doses received by the patients in the NAT group and FOLFIRINOX–NAT group was the same (6 (range, 2–8)). Other pathological features noted in the resected samples are detailed in Table 1.

### 2.2. MUC5AC Detection and Distribution for Patients with Resected PDA

The MUC5AC detection/distribution among the samples is presented in Appendix A. Both glycoforms (45M1 and CLH2) were present in 96% of the tested samples, and there was 100% overlap between them, i.e., all the samples positive for CLH2 were also positive for 45M1. Four samples in the NAT group were negative for both. EC-45M1 was detected in 72% of the tumors, and all of these tumors had 45M1 and CLH2 expression. CLH2 (IM) was detected only in the cytoplasm and not in the apical or EC regions (Appendix A). The mature glycoform, 45M1, was detected only in the apical and EC regions. The mean expression levels (H-scores) for 45M1 and CLH2 were 148.5 and 145.27, respectively. A total of 25% (24/96 and 14/43 in NAT group, and 10/57 in Ups group) of PDAs that expressed 45M1 did not have EC-45M1 detected. 

A simple linear regression analysis showed a significant (*p* < 0.001) positive correlation between CLH2 and 45M1 for all populations (NAT, UpS, and NAT + UpS). A simple logistic regression showed a significant correlation (*p* < 0.001) between EC-45M1 detection and CLH2 or 45M1 expression, and the mean expression levels (H-score) were 2–3 times greater in EC-positive tumors than in EC-negative tumors (Appendix A). The mean CLH2 expression and EC-45M1 detection rates were significantly lower in the NAT group than in the UpS group (Figure 1A). 45M1 expression was lower, with a trend toward significance in the NAT group (Table 2). We did not have enough Gem-based therapy patients (*n* = 5) in the NAT group to learn if MUC5AC imparts more resistance to 5FU or Gem-based therapy.

### 2.3. MUC5AC Expression Levels and Pathological Treatment Responses in PDA Post-NAT

We first studied the effect of MUC5AC on the pTR for the entire population (*n* = 100) and performed a sub-group analysis on the NAT group. For the first analysis, we had three groups: objective response (OR) vs. no response (NR) vs. UpS. Here, the UpS population served as a control, as it represented the treatment-naïve population. We compared the mean expression of 45M1 and CLH2 and EC detection rates among the groups (Table 2). 

The mean CLH2 expression differed significantly, and 45M1 had a strong trend toward significance (*p* = 0.06) among the three groups (Figure 1B). The OR and UpS groups had the lowest and highest expression levels for both MUC5AC glycoforms (Table 2). The NR group had levels close to those of the UpS group and higher than those for the OR group. In head-to-head comparisons, MUC5AC expression (mean 45M1 and CLH2) was significantly lower in the OR groups than in the UpS groups; NR vs. OR and NR vs. UpS were not significantly different. 

The proportion of EC-45M1 detection (positive vs. negative) differed among the three groups (Table 2). More subjects with OR (44%) were EC-45M1-negative than NR (37%) and UpS (18%) subjects. In the head-to-head comparisons, EC-45M1 detection significantly differed, following the same trend as 45M1. The difference between OR and UpS was significant; the others were not. The detection rates for the NR group were closer to those for the OR group than those for the UpS group, although MUC5AC expression was closer to that of the UpS group than that for the OR group. 

For the NAT group, the mean H-scores for 45M1 and CLH2 were 120 (30–210) and 100 (30–210), respectively. The lower CLH2 expression group (≤100, *n* = 22) had a significantly (*p* = 0.04) higher percentage of patients with OR (77% vs. 48%) than the higher CLH2 expression group (>100, *n* = 21) as discussed in Table 3 below. There was a trend toward significance (*p* = 0.07) for 45M1 (≤100 (*n* = 22) vs. >100 (*n* = 21), 76% vs. 50%) for the same cut-off value (H-score = 100). EC detection was significantly lower (*p* < 0.001) for lower CLH2 (≤100 vs. >100, 37% vs. 90%) and 45M1 (24% vs. 91%). This evidence suggests that a threshold of 100 for H-score (for CLH2 > 45M1) could be used as a cut-off to identify patients who respond to therapy. The FOLFIRINOX–NAT sub-group group (*n* = 36) had the same trend, but the *p* values were only 0.07 for CLH2 and 0.08 for 45M1.

In this study population, the UpS group had significantly better median PFS (313 days in NAT vs. 441 days in UpS, *p* = 0.003) and median OS (741 days vs. 995 days, *p* = 0.0016) than the NAT group (Appendix A).

### 2.4. Impact of MUC5AC on Pathological Features of Resected PDA

We examined the impact of MUC5AC via logistic regression analysis (Table 4) for all the patients in the study, irrespective of NAT (*n* = 100). 

The pTR (OR vs. NR vs. UpS) was affected by 45M1 and CLH2 levels and EC-45M1 detection. CLH2 expression influenced tumor size (≤2 cms vs. >2 cms), and EC-45M1 detection influenced NAT and association with premalignant lesions (AwPml) such as intraductal papillary mucinous neoplasms (IPMN) and pancreatic intraepithelial neoplasia (PanIN). A trend toward significance was noted in the effect of 45M1 on tumor size, AwPml, and NAT, and CLH2 on AwPml. 

A similar analysis for the UpS group (*n* = 57) gave information about the role of MUC5AC on treatment-naïve PDAs. For this population, 45M1 and CLH2 expression were not associated with other pathological features. For the NAT group (*n* = 43), 45M1 expression levels were significantly related to incomplete resection (residual disease, R0 vs. R1-R2) and perineural invasion (PNI), but EC-45M1 detection was related to AwPml (Appendix A). An encouraging trend toward significance was noted in the effect of 45M1 on margin status (Ms), CLH2, EC-45M1 detection, and EC-45M1 composite score (CS) for residual disease and PNI. In the FOLFIRINOX group (*n* = 36), 45M1 and CLH2 affected residual disease, Ms, and PNI. A trend toward significance was noted for the effect of EC-451 detection on AwPml and residual disease and EC-45M1 CS on residual disease and PNI.

### 2.5. The Impact of MUC5AC on Progression-Free Survival and Overall Survival in R-PDA

The study population was a heterogeneous group with multiple variables (NAT, UpS, NAT-CRT, and various AT combinations). We minimized heterogeneity by assessing the relationship of MUC5AC with various groups, such as the NAT group, patients receiving FOLFIRINOX in the NAT group, patients receiving Gem-based AT (Gem-only, Gem/Cap, and Gem-NP), and Gem-only AT in the UpS group. 

We first analyzed the impact of MUC5AC expression on the NAT group (*n* = 43) and on progression-free survival (PFS) and overall survival (OS) via univariate analysis (UVA) and multivariate analysis (MVA) (Table 5 and Table 6). The median PFS and OS (in days) of this group were 313 (CI = 212, 391) and 741 (CI = 452, 857), respectively. According to the UVA, there was no significant relationship (*p* > 0.05) between MUC5AC expression and PFS or OS.

For the resected specimens, EC-45M1-negative had a better PFS according to the MVA (Hazard Ratio (HR): 0.2, 95% CI: 0.059–0.766, *p* = 0.01). 45M1 expression was significant (*p* = 0.0398) according to the MVA for PFS, but the HR (0.96) was not impressive enough to make it a clinically significant factor. Other features that affected PFS were pathological grade (G1-G2 vs. G3), Ms, NAT-CRT, site of recurrence, OS pathological grade (G1-G2 vs. G3), and Ms. Among the 43 patients in this group, 16 (37%) did not receive adjuvant therapy, and Gem-only or Gem-based therapy was given to the others (10 Gem-only, 3 Gem-NP, 4 Gem/Cap, 7 FOLFIRINOX, 1 FOLFOX, 1 FOLFIRI, and 15 FU-only). The CLH2 expression level did not significantly affect PFS (*p* = 0.07 and HR of 1.02). According to the MVA, 45M1 and CLH2 expression impacted OS, with HRs of 0.959 and 1.043, respectively (Table 6). EC-45M1 detection did not significantly affect OS. The pathological grade and Ms were also factors that significantly influenced OS. 

A similar trend was evident for patients in the NAT group treated with FOLFIRINOX (*n* = 36). The median PFS and OS (in days) of this group were 336 (CI = 162, 599) and 741 (CI = 334, 1003). The results are summarized (Table 6) as follows: (a) MUC5AC did not change OS or PFS according to the UVA; (b) EC-45M1 detection was a significant factor (HR: 0.2, 95% CI: 0.052–0.847, *p* = 0.02) for PFS according to the MVA (Appendix A); (c) 45M1 and CLH2 expression was significant, but the HRs were clinically not impressive (45M1—0.9 and IM—1.02); (d) lymphovascular invasion (LVI), Ms, Awpml, and site of recurrence were other pathological factors significantly affecting PFS according to the MVA; and (e) the MVA showed that the factors affecting OS were (Appendix A) 45M1 and CLH2 (HRs of 0.86 and 1.12, respectively), peri-pancreatic soft tissue extension (PPE), pathological grade, LVI, Ms, tumor size, AwPml, and site of recurrence. 

UpS patients who received Gem-based AT (*n* = 51, 33 Gem-only, 11 Gem/Cap, and 7 Gem-NP) had notable results (Appendix A). 45M1 positively impacted PFS only (not OS) according to the MVA (HR of 1.03, *p* = 0.03). There was a trend toward significance for the effect of CLH2 (HR of 0.97, *p* = 0.07) on PFS. EC-45M1 did not impact PFS or OS. Other factors influencing PFS and OS are shown in Appendix A. MUC5AC did not change the outcomes in the other groups (UpS (*n* = 57), Gem-only (*n* = 33), or all patients (*n* = 100)). 

The impact of MUC5AC in the resected samples post-NAT are summarized as follows: (i) EC-45M1 detection post-NAT increased the risk of recurrence; (ii) MM (45M1) expression negatively impacted PFS and OS, but the HRs were not clinically significant; (iii) IM (CLH2) expression had a positive effect on OS; (iv) for the UpS group, MUC5AC expression did not influence the outcomes, but MM (45M1) had a minimal positive effect on PFS in a sub-group that received Gem-based therapy. 

### 2.6. MUC5AC Expression in Primary and Distant Metastatic Sites

In the study population, 10 patients had metastatic disease. One patient with micrometastasis of the liver identified during primary resection was excluded from this comparison (Appendix A). Four out of nine patients (treatment-naïve) received no systemic therapy before the biopsy. The others (*n* = 5) received systemic therapy (2-FOLFIRNIX, 1-FOLFOX, 1-Gem/nab-paclitaxel, 1-Gem only). All nine biopsies were positive for 45M1, EC-45M1, and CLH2. The mean expressions of 45M1, EC-45M1 CS, and CLH2 were 199, 199, and 196, respectively. Among the metastatic sites, the lungs had the highest H-scores, and the peritoneum had the lowest (lung (300) > other organs such as ovary, small bowel, and bladder (260) > liver (170) > primary tumor > peritoneum (100)). The 45M1 and CLH2 H scores were similar. The numbers of these two groups were too low to determine the effect of the treatment on MUC5AC expression. When MUC5AC expression was compared between metastatic sites (*n* = 9) and resected specimens (*n* = 100), the CLH2 and EC-45M1 H-scores were significantly higher for metastatic sites than for primary tumors (Appendix A). The distributions (percentages of positive cells) of 45M1 and CLH2 were significantly higher for metastatic sites. We compared MUC5AC expression in the treatment-naïve (*n* = 4) and NAT groups and the treated (*n* = 5) and UpS groups (Appendix A). There was a similar trend for these comparisons.

## 3. Discussion

Our study provided insight into the clinical significance of MUC5AC glycoforms in PDA and showed an association of pTR with MUC5AC glycoforms. We identified the specific population (post-systemic therapy, NAT group in this study) and the location of MUC5AC detection (EC) that need to be explored to understand its impact on PDA. Our study reaffirms the association of MUC5AC expression with malignancy, as it is detected in all treatment-naïve patients (UpS group). The detection rates were higher than reported in other studies, including our previous TMA study [21,34,35,36,37,38]. The 100% concordance and positive correlation between 45M1 and CLH2 and detected MUC5AC supports the theory that IM matures through glycosylation, moves to the apical region, and is released into the EC region [28]. 

The inclusion of treatment-naïve (UpS) and treated (NAT) patients allowed us to examine, more effectively than in previously reported studies, the influence of treatment on MUC5AC expression and vice versa (Table 2 and Appendix A). MUC5AC expression was impacted by NAT and pTR (Figure 1 and Table 4), indicating its possible role in influencing the pathways or mechanisms of action of chemotherapeutic drugs. The countereffects of CLH2 and 45M1 for the NAT group (Table 4 and Table 5, Appendix A) and of PFS in the Gem-based therapy sub-group of UpS (Appendix A) need to be investigated further to determine if MUC5AC expression can help in treatment selection. For most preclinical studies, the association of MUC5AC and Gem resistance was the focus of the investigations, and some potential pathways, such as the MUC5AC/β-catenin/c-Myc axis, were identified [39,40,41,42]. Our study did not have enough patients in the NAT group on Gem-based therapy (*n* = 5) to make substantial conclusions, but MUC5AC expression could be drug-specific (Appendix A). The impact difference in the HRs of intracellular MM (45M1) in the NAT group (HR of 0.96) and in the FOLFIRNOX sub-group (HR of 0.86) strengthens this hypothesis, which still needs further investigation.

The baseline MUC5AC expression did not influence the major pathological features or outcomes (OS and PFS) based on the treatment-naïve patients (UpS group). This, along with the differential impact of CLH2 (positive impact on NAT group, no impact on UpS group, negative impact on Gem-based adjuvant therapy subgroup), explains the mixed results for previous IM-only-based studies of this population [34,35,36]. Alternatively, post-NAT MUC5AC expression and EC detection impacted the pathological features (such as pTR, incomplete resection, Ms, and PNI) and outcomes (PFS). The small difference in EC-MUC5AC detection rates between the NR (63%) and OR (56%) groups, despite having larger differences in intracellular MUC5AC expression (mean H-score, OR vs. NR = 113 vs. 153) for patients who had curative resection without progression or distant metastasis, highlights the clinical significance of EC-MUC5AC in PDA, as demonstrated in preclinical studies [39]. The number of NAT doses (median of six) is similar to that in clinical practice, as reflected in most clinical trials (NORPACT, ESPAC-5, and SWOG 1505) [13,43,44]. We did not have enough samples to further explore the differences between primary and metastatic lesions. However, metastatic lesions (*n* = 9) had significantly higher MUC5AC expression and a 100% EC-45M1 detection rate, implying that it has a role in disease progression and distant metastasis.

Prospective studies with a large sample size should validate the findings of this retrospective study. Surgical resection was a confounding factor for survival analysis for both groups (NAT and UpS). Post-operative management was not uniform in either of the groups. One-third of the NAT group patients did not receive AT. For the rest, Gem-only or Gem-based therapy was administered. Most of the NAT group received Gem-only (33/57) or Gem-based (51/57) treatment, which is presently not the standard of care. The use of NAT was not popular during the early part of the study period. There could have been selection bias in administering NAT. We do not have MUC5AC expression in the recurrent lesions to confirm that it was similar to that of the resected samples. NAT-CRT is another confounding factor in assessing the association between pTR and MUC5AC expression. We did not have a large enough sample size to make substantial conclusions for NAT or AT selection (FOLFIRINOX vs. Gem-NP). Crucial factors like tumor location, resectability status at diagnosis, type of surgery, and changes in CA19-9 levels were not included in the multivariate model. Future studies should incorporate these variables to obtain more conclusive results.

## 4. Materials and Methods

### 4.1. Study Design and Population

This study was conducted at the Ohio State University Comprehensive Cancer Center (OSU-CCC), Columbus, Ohio, after receiving appropriate approval from the Office of the Institutional Review Board. The Total Cancer Care Program (TCCP), a division of the OSUCCC, identified the patients who underwent resection for PDA at this institution and provided archived clinical and pathologic data. They also provided an H & E sections and multiple unstained sections of representative primary tumor formalin-fixed paraffin-embedded (FFPE) blocks from the study period, January 2010 to June 2021. Any additional clinical or pathologic information was collected through a review of medical records by the principal investigator (AM).

### 4.2. Immunohistochemistry

Immunohistochemical detection of MUC5AC isoforms was accomplished using mouse monoclonal antibodies CLH2 (NBP2-44455, NovoBiologicals, CO, USA) and 45M1 (ab3649, Abcam, MA, USA) on an Agilent DAKO autostainer link 48 system (Agilent Technologies, Santa Clara, CA, USA). Formalin-fixed paraffin-embedded tissue sections were deparaffinized/rehydrated, and antigen retrieval was performed with Agilent DAKO target retrieval solution with citrate buffer (pH 6.1, S169984-2; Agilent Technologies, Santa Clara, CA, USA) at 95 °C for 27 min. For CLH2, the primary antibody was incubated at a dilution of 1:800 for 30 min at room temperature. The primary antibody was detected using a VECTASTAIN^®^ Elite^®^ ABC Universal PLUS Kit (PK-8200; Vector laboratories, Newark, CA, USA) with a polymer peroxidase diaminobenzidine chromogen system. For 45M1, the primary antibody was incubated at a dilution of 1:1000 for 30 min at room temperature. It was detected using an Agilent Envision Flex Kit (K80002, Agilent Technologies, Santa Clara, CA, USA) and a polymer peroxidase diaminobenzidine chromogen system.

A pathologist (AE) with expertise in pancreaticobiliary pathology reviewed the immunostains. The intracellular localizations for the IM (CLH2) and MM (45M1) isoforms, including the cytoplasmic and apical expression levels, were determined. Individual tissue sections were scored for the percentage of reactive tumor cells and intensity of reactivity (0, no staining; 1+ weak staining; 2+ moderately intense staining; and 3+ strong staining). The H-score was derived from the product of the percentage of tumor cells and intensity of staining, resulting in a value between 0 and 300 [45]. Also, the presence or absence of extracellular (EC-45M1) expression for MM was noted.

### 4.3. Clinical and Pathology Data Collection

This study was conducted in compliance with all the applicable institutional ethical guidelines for the care, welfare, and use of animals. Demographic data, such as age, gender, race, social history (smoking and alcohol use), and laboratory information (CA 19-9 and total bilirubin) were collected. Clinical information regarding the date of diagnosis, clinical staging, type of surgery, residual disease (R0 vs. R1/R2), neoadjuvant and adjuvant therapy received, and performance status were collected from the medical records.

Pathology data were obtained from the pathology reports of the definitive surgical resection specimens. It included the histological grade of the tumor (well differentiated (Grade 1), moderately differentiated (Grade 2), and poorly differentiated (Grade 3)), Ms, tumor size (≤2 cm (cms) vs. >2 cms), LVI, perineural invasion (PNI), perivascular invasion (PVI), peri-pancreatic soft tissue extension (PPE), lymph node (LN) metastasis, and the presence of any reported pre-malignant lesions such as PanIN or IPMN and distant metastasis.

For 70 patients, the tumor (T) staging had to be reclassified as there was a significant change in the TNM staging from the prior American Joint Committee on Cancer (AJCC) 7th edition to the currently used AJCC 8th edition. The pathological treatment response (pTR) noted on the resected samples for patients who received NAT was classified into two major groups: OR and NR. OR included patients with near a complete response (nCR) and a partial response (PR). nCR is characterized by single cells or a small group of cancer cells, corresponding to a tumor regression score of 1 according to the modified Ryan scheme for tumor regression score (TRS) [46]. PR was characterized by residual cancer with evident tumor regression, but with more than single cells or rare small groups of cancer cells (TRS 2). The NR group included patients with extensive tumors with no evident tumor regression (TRS 3).

### 4.4. Statistical Considerations

Descriptive statistics were used to summarize patient baseline characteristics. ANOVA models or Chi-square tests were applied to compare MUC5AC expression between treatment response groups. Kaplan–Meier curves for progression-free survival and overall survival were used for comparing patient survival between the NAT and UpS groups. Log-rank tests were applied for testing the differences between two survival curves. Univariate logistic regression models were employed to assess the association between pathological features and MUC5AC expression. PFS (date of diagnosis to recurrence) and OS (date of diagnosis to death or last day of follow up) analyses were performed through Kaplan–Meier curves and univariate and multivariate Cox regression models over pathological features. Bonferroni correction was applied for *p*-value adjustments. All analyses used SAS 9.4 (SAS, Cary, NC, USA).

## 5. Conclusions

Our study provides evidence suggesting the clinical value of MUC5AC expression in resected PDA, specifically for post-NAT patients. Our hypothesis that MM detected in the apical and EC regions influences the outcome more than IM was confirmed. We showed that MM, which was not studied earlier, has clinical utility (pTR and outcomes). This association has implications for deciding the duration of NAT and the timing of the surgical resection. Based on the findings reported in this study and in others, the potential clinical implications, pending confirmation in large prospective trials, of MUC5AC tissue testing could help in NAT management as follows: a) the drop in tumor intracellular MM expression and the conversion of EC-MUC5AC from positive to negative may indicate a response to NAT and low risk for recurrence, suggesting an ideal time for surgery; b) stable intracellular MM and the conversion of intracellular MUC5AC-from positive to negative could indicate a poor response to NAT but still a low risk of recurrence; and c) stable intracellular MM and persistent EC-MUC5AC detection could indicate a poor response to NAT and a high risk of recurrence.

Future research should focus on elucidating the underlying mechanisms linking MUC5AC expression to the treatment outcomes of post-NAT PDA patients. Furthermore, the failure of NPC-1C antibody treatment to improve the outcome in a recent clinical trial strengthens the importance of identifying the appropriate MUC5AC glycoform for therapeutic purposes [47]. Thus, these findings will help clinicians consider other treatment options, including changing the NAT regimen, proceeding to surgery, or using CRT, especially in the last two scenarios.

## 6. Patents

MUC5AC in pancreatic adenocarcinoma (PCT/US2024/012771, 24 January 2024)) [patent pending].

## Figures and Tables

**Figure 1 ijms-25-09041-f001:**
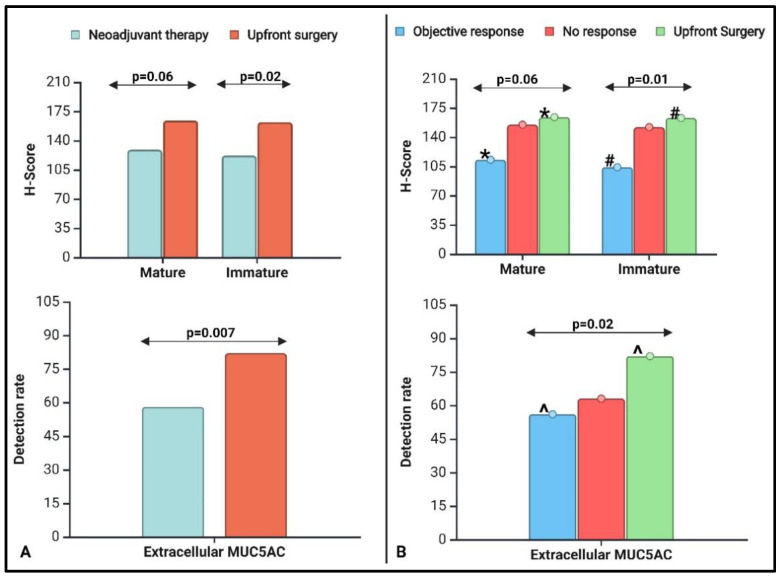
Comparison of MUC5AC glycoforms (mean H-scores) among the tested groups, (**A**) neoadjuvant therapy group vs. upfront surgery (UpS) and (**B**) objective response (OR) vs. no-response vs. UpS; * OR vs. UpS—*p* = 0.04; # OR vs. UpS—*p* = 0.01; ^ OR vs. UpS—*p* = 0.01.

**Table 1 ijms-25-09041-t001:** Pathological features identified on the resected pancreatic tissues (*n* = 100).

Pathological Feature	Distribution in Percentages (%)
Differentiation (G1 vs. G2 vs. G3)	11 vs. 62 vs. 27
Peripancreatic extension	50
Treatment effect noted in NAT group (*n* = 43). Objective response (OR) vs. no response (NR).	27 (OR) vs. 16 (NR) (OR: nCR—4; PR—23)
T-stage, T1 vs. T2 vs. T3 vs. T4	23 vs. 54 vs. 21 vs. 2
Tumor size, ≤2 cms vs. 2–4 cms vs. >4 cms	23 vs. 55 vs. 22
Lymph vascular invasion identified	64
Perineural invasion identified	80
Margins positive	29
Residual disease: R0 vs. R1 vs. R2	72 vs. 26 vs. 2
Node positivity: N0 vs. N1–2	29 vs. 71
Association with premalignant lesion: PanIN vs. IPMN vs. no lesion identified	54 vs. 9 vs. 37

G—grade; NAT—neoadjuvant therapy; cms—centimeters; nCR—near complete response; PR—partial response; PanIN—pancreatic intraepithelial neoplasia; IPMN—intraductal papillary mucinous neoplasms.

**Table 2 ijms-25-09041-t002:** Comparison of 45M1 and CLH2 among OR vs. NR vs. UpS groups (*n* = 100).

	Mean H-Scores OR vs. NR vs. UpS	Head-to-Head Comparisons
45M1 *	113.3 vs. 154.38 vs. 163.6 (*p* = 0.0612)	OR vs. UpS (*p* = 0.0498)
		NR vs. UpS (*p* = 0.9308)
		OR vs. NR (*p* = 0.3245)
CLH2 *	104.07 vs. 151.88 vs. 162.93 (*p* = 0.0184)	OR vs. UpS (*p* = 0.0140)
		NR vs. UpS (*p* = 0.8969)
		OR vs. NR (*p* = 0.2015)
EC-45M1 ^#^	56% vs. 63% vs. 82% (*p* = 0.002)	OR vs. UpS (*p* = 0.0155)
		NR vs. UpS (*p* = 0.1008)
		OR vs. NR (*p* = 0.7548)

OR—objective response; NR—no response; UpS—upfront surgery; * H-score; # detection rate.

**Table 3 ijms-25-09041-t003:** Impact of MUC5AC expression on response and extracellular MUC5AC detection.

MUC5AC Glycoform	Expression in H-Score	Impact on Response			Impact on EC-45M1	
		OR (%)	NR (%)	*p*-Value	Detection (%) *	*p*-Value
CLH2	≤100	77	23	0.04	37	<0.01
	>100	48	52		90	
45M1	≤100	76	24	0.07	24	<0.01
	>100	50	50		91	

OR—objective response; NR—no response; * positivity rate.

**Table 4 ijms-25-09041-t004:** Univariate logistic regression model for MUC5AC expression in all the patients (*n* = 100).

Pathological Feature	45M1	CLH2	EC-45M1
Pathological differentiation, G1–2 vs. G3	0.6866	0.8559	0.4358
Peripancreatic extension	0.6200	0.8741	0.1841
**Treatment effect, OR vs. NR vs. UpS**	**0.0317**	**0.0112**	**0.0081**
Lymphovascular invasion	0.8954	0.9811	0.3744
Perineural invasion	0.4129	0.4802	0.8238
Margins	0.5935	0.8658	0.5832
R0 vs. R1-R2	0.2971	0.6141	0.3642
**Tumor size (≤2 cms vs. >2 cm)**	0.0523	**0.0478**	0.1796
N0 vs. N1–N1	0.9004	0.8697	0.5832
**Premalignant lesion, yes vs. no**	0.0585	0.0643	**0.0007**
Neoadjuvant CRT	0.3952	0.2876	0.3592
**Neoadjuvant therapy**	0.0612	**0.0264**	**0.0088**
Site of recurrence (none vs. local vs. distant)	0.1381	0.1405	0.6321

G—grade; OR—objective response; NR—no response; UpS—upfront surgery; EC—extracellular; CRT—chemoradiation.

**Table 5 ijms-25-09041-t005:** Multivariate analysis of clinicopathological features for progression-free survival of neoadjuvant group (*n* = 43).

Parameter		Pr > ChiSq	Hazard Ratio	95% Wald Confidence Limits
**45M1—H-score**		**0.0398**	**0.968**	**0.938–0.998**
**CLH2—H-score**		0.0741	1.029	0.007–1.063
**EC detection**	**Negative**	**0.0178**	**0.213**	**0.05–0.766**
**Pathological differentiation,** **G1–2 vs. G3**	**G1-2**	**0.0385**	**0.350**	**0.130–0.946**
Peripancreatic extension	NO	0.5607	0.734	
**Treatment effect, OR vs. NR**	OR ^1^	0.7106	0.789	0.226–2.755
*Lymphovascular invasion*	*NO*	0.0551	0.313	0.095–1.026
Perineural invasion	NO	0.6831	0.718	
**Margins**	**Negative**	**0.0014**	**0.127**	**0.03–0.452**
R0 vs. R1–R2	R0	0.7909	0.816	
Tumor size (≤2 cms vs. >2 cm)	≤2 cms	0.6069	1.562	
N0 vs. N1-N1	N0	0.7950	0.853	
Premalignant	None	0.9336	0.952	
**NAT CRT**	**Had NAT CRT**	**0.0459**	**0.369**	**0.138–0.982**
**Site of recurrence**	**Distant metastasis ^1^**	**0.0379**	**4.838**	**1.092–21.433**
**Site of recurrence**	**Local recurrence ^1^**	**0.0176**	**8.798**	**1.462–52.965**

1—vs. no recurrence; G—grade; OR—objective response; NR—no response; UpS—upfront surgery; EC-extracellular; CS—composite score; CRT—chemoradiation; NAT—neoadjuvant.

**Table 6 ijms-25-09041-t006:** Multivariate analysis of clinicopathological features for overall survival of neoadjuvant group (*n* = 43).

Parameter		Pr > ChiSq	Hazard Ratio	95% Wald Confidence Limits
**45M1—H-score**		**0.0170**	**0.959**	**0.926–0.993**
**CLH2—H-score**		**0.0205**	**1.043**	**1.006–1.080**
EC detection	Negative	0.1406	0.343	
**Pathological differentiation,** **G1–2 vs. G3**	**G1-2**	**0.0038**	**0.193**	**0.06–0.588**
Peripancreatic extension	NO	0.5877	1.294	
Treatment effect, OR vs. NR	OR	0.7734	1.219	
Lymphovascular invasion	NO	0.1567	0.405	
Perineural invasion	NO	0.5547	0.609	
**Margins**	**Negative**	**<0001**	**0.040**	**0.009–0.180**
*R0 vs*. *R1-R2*	*R0*	*0.0878*	*3.986*	*0.815–19.493*
Tumor size (≤2 cms vs. >2 cm)	≤2 cms	0.1407	3.845	
N0 vs. N1-N1	N0	0.2046	0.429	
Premalignant	None	0.3680	1.670	
NAT CRT	Had NAT CRT	0.1987	0.542	
Site of recurrence	Distant metastasis ^1^	0.8868	0.894	
Site of recurrence	Local recurrence ^1^	0.5050	1.961	

^1^—vs. no recurrence; G—grade; OR—objective response; NR—no response; EC—extracellular; CS—composite score; CRT—chemoradiation; NAT—neoadjuvant therapy.

## Data Availability

For ethical reasons, the data presented in this study are available upon request from the corresponding author.

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
