# Peer review of "Mature MUC5AC Expression in Resected Pancreatic Ductal Adenocarcinoma Predicts Treatment Response and Outcomes"

_ijms, 2024, doi:10.3390/ijms25169041_

Round 1

Reviewer 1 Report

Comments and Suggestions for Authors

Manne et al investigated the clinical significance of two major MUC5AC glycoforms, IM and MM, in resected PDA specimens. The authors concluded that the MM expressions were associated with treatment response and outcomes in the NAT group. Although this manuscript has some important findings, I have some questions and comments about this manuscript.

1.     NAT is widely used to treat PDA. In this study, MUC5AC expression was impacted by NAT. Therefore, the difference between OR and UpS was significant in Table 2. It may be useful to evaluate a response to NAT. It would be better to show Kaplan-Meier curves (PFS and OS) according to NAT(OR, NR), and UpS.

      PFS and OS should be changed by tumor markers, resectability status, tumor location and surgical procedures. Many studies have reported that normalization or decrease of tumor markers such as CA19-9 is predictive marker for treatment response. How about these factors in this study?

2.     The authors mentioned that EC-45M1 detection post-NAT is a poor prognostic marker in line 309. However, EC detection post-NAT did not significantly affect the OS on MVA. Please explain.

3.     MM is identified by 45M1, 1-13M1, 9-13, and 2-11M1 mAbs in line 79-80. In this study, only 45M1 was used. Please explain why you selected 45M1 mAb.

4.     This study includes relative long period. NAT should be recently introduced. Is there any difference in formalin fixation time? Please explain the formalin-fixation duration of resected PDA specimens on IHC staining.

5.     Multivariate analyses were performed in Table 4 and 5. Please show the cut-off values of 45M1-H-score and CLH-H-score.

6.     How do you define PFS and OS? From NAT or Surgery?

7.     Please add the Figure 1.

8.     Please add the figure or Table about line 230-239.

9.     In Table 1, total percentage of node positivity is 91 (20+71), not 100. Please confirm. 

10.  Many abbreviations are not explained. Please, look through the manuscript, and write out the full term for abbreviations such as OS, PFS, Ms, CS, and UVA at its first use.

Author Response

Please find the reply attached. 

Reviewer 2 Report

Comments and Suggestions for Authors

Manne et al conducted a retrospective study to evaluate the prognostic and predictive value of two major glycoforms of mucin 5AC. The article is well-written and interesting, given the critical need for predictive biomarkers in pancreatic cancer

However, as the authors mentioned, the study population was heterogeneous (pre and postoperative management were not uniform), thus the results should be interpreted cautiously. Validation in larger, prospective trials is needed.

I have a question regarding the conclusions of the study. The authors stated that: "the potential clinical implications of MUC5AC tissue testing could help NAT management as follows: a) the drop in tumor intracellular MM expression and conversion of EC-MUC5AC  positive to negative may indicate a response to NAT and low-risk for recurrence, suggesting an ideal time for surgery; b) stable intracellular MM and conversion of intracellular MUC5AC-positive to -negative could indicate a poor response to NAT but still low-risk of recurrence; and c) stable intracellular MM and persistence EC-MUC5AC detection could indicate poor response to NAT and high-risk of recurrence".

How are these conclusions supported by the results of the study? For example, to assess  the drop in tumor intracellular MM expression and conversion of EC-MUC5AC  positive to negative, we would also need to have a baseline detection of MUC2 (before and after NAT).

Also, can MUC5AC expression be evaluated on EUS-FNA/FNB samples?

Author Response

Please find the reply attached. 

Round 2

Reviewer 1 Report

Comments and Suggestions for Authors

The authors have responded appropriately.